# Anxiety, Depression, and PTSD among College Students in the Post-COVID-19 Era: A Cross-Sectional Study

**DOI:** 10.3390/brainsci12111553

**Published:** 2022-11-15

**Authors:** Xing Wang, Nan Zhang, Changqin Pu, Yunyue Li, Hongguang Chen, Mengqian Li

**Affiliations:** 1School of Life Sciences, Nanchang University, Nanchang 330036, China; 2Clinical Medical Experiment Center, Nanchang University, Nanchang 330036, China; 3Department of Psychology, School of Public Policy and Administration, Nanchang University, Nanchang 330031, China; 4Queen Mary College, Nanchang University, Nanchang 330006, China; 5Peking University Sixth Hospital, Peking University Institute of Mental Health, NHC Key Laboratory of Mental Health (Peking University), National Clinical Research Center for Mental Disorders (Peking University Sixth Hospital), Beijing 100083, China; 6Department of Psychosomatic Medicine, Gao Xin Hospital of the First Affiliated Hospital of Nanchang University, Nanchang 330029, China

**Keywords:** COVID-19, mental health, anxiety, depression, PTSD, college students

## Abstract

In the post-COVID-19 era, significant changes have taken place regarding the epidemic, the economy, family and social life. However, it remains unclear how these changes encompass the psychological symptoms of college students. We carried out a cross-sectional study to investigate anxiety, depression, and post-traumatic stress disorder (PTSD) symptoms among college students from 10 November 2020, to 16 November 2020. The questionnaire included a self-designed canvas, Generalized Anxiety Disorder 7 (GAD-7), Patient Health Questionnaire 9 (PHQ-9), and Impact of Event Scale (IES-R). Factors associated with psychological symptoms were estimated by ordered and non-conditional logistic regression analysis. Of 4754 participants, 25.0%, 29.7%, 3.4%, 15.3%, 17.1%, and 2.9% reported anxiety, depression, PTSD symptoms, one, any two, and all three, respectively. In cases with anxiety or depression symptoms, there was a 9.11% comorbidity with PTSD. Factors associated with fears of being infected, social, family, and economic changes increased the risk of psychological symptoms in college students caused by COVID-19. Female college students, identified with anxiety or depression symptoms, were at a lower risk of developing PTSD symptoms (OR, 0.61, 95% CI: 0.43–0.86). Non-medical majors at university, rural residence, higher educational background, fear of taking public transport, and deterioration of family relationships increased the risk for PTSD symptoms among male respondents with anxiety or depression symptoms due to COVID-19. Factors correlated with psychological symptoms had expanded from the fear of being infected to extensive social, family, and economic changes caused by COVID-19. Therefore, screening and interventions for psychological symptoms should be consistently strengthened and more targeted to college students in the post-COVID-19 era.

## 1. Introduction

Coronavirus disease 2019 (COVID-19) was declared a pandemic by the World Health Organization (WHO) on 11 March 2020 [1,2,3]. As of 13 February 2022, more than 404.9 million people have been infected with COVID-19, and 5.7 million deaths have been reported globally [4]. It is reported that public health emergencies like COVID-19 may lead to psychological and physical damage to the public, and that the psychological impact caused by such events will last for a long time [5]. Relative to other social groups, college students are more gullible regarding the COVID-19 epidemic, experiencing uncertainty and sudden semester interruptions.

Previous studies focused on the psychological conditions of college students at the beginning of the pandemic. Most studies reported higher anxiety and depression symptoms during the pandemic outbreak [6,7,8,9]. Although the early stages of the epidemic were more heavily researched, not enough effort has been devoted to research into the post-COVID-19 era, relatively. Studies on the mental health of college students in the post-COVID-19 era have mainly involved exploring the psychological status, such as the evaluation of anxiety, depression, PTSD symptoms, insomnia, stress, and fatigue level, as well as obtaining a picture of their coping styles, quality of life, social support, psychological resilience, etc. [10,11,12]. Nevertheless, the association between mental health status and the factors influenced by several different COVID-19 dimensions among college students has not been adequately explored in depth.

In the post-epidemic period, although the physical health of the population is gradually recovering, the adverse mental health consequences of the pandemic may persist and even worsen [13]. China is one of the countries that entered the post-epidemic era in the early stages, and its social economy has also been impacted by the epidemic. Meanwhile, social activities, lifestyles, family relationships, and learning environments will also change along with the development of this epidemic [14,15]. How will these changes affect the mental health of college students? This study attempted to explore the prevalence of anxiety, depression, and PTSD symptoms, particularly in relation to socioeconomic changes, with the aim of exploring the four areas: (a) College students continued to experience high levels of anxiety, depression, and PTSD symptoms relative to the results of our laboratory studies at the time of the outbreak [16]. (b) Family relationships were associated with the presence of these three symptoms. (c) The psychological state of college students was influenced by changes in family economic status due to the COVID-19 epidemic. (d) In terms of gender, females in particular were more likely to exhibit a correlation with these three symptoms.

## 2. Materials and Methods

### 2.1. Study Design

This cross-sectional study was conducted by using non-probability sampling from 10 to 16 November 2020, when the pandemic of COVID-19 had been brought under control and most college students had returned to regular school life. The participants in this study were college students from Jiangxi Province, China. The online questionnaire with a quick response code (QR code), which was administered through a web-based platform, was sent to colleges located in Jiangxi. A total of 4754 pieces of poll data were collected and further analyzed. Electronic informed written consent was obtained from all respondents before data collection. Moreover, the Research Ethics Board approved the scrutiny at the First Affiliated Hospital of Nanchang University (approval number: 2020-050).

### 2.2. Measurements

The measurements consisted of four sections: sociodemographic data, self-reported COVID-19-related data, Generalized Anxiety Disorder 7-item Scale, Patient Health Questionnaire-9, and Impact of Event Scale.

### 2.3. Sociodemographic Data and COVID-19-Related Data

The sociodemographic data embodied gender, age, education (whether they are medical students or not), census registration, and household income level (low/middle/high). COVID-19-related data included personal protective behaviors during the post-pandemic period as “Whether catching a cold within one week of the survey”, “Whether relatives and friends are infected with COVID-19”, “Worried about whether COVID-19 is widely spread again in China”, “Worried about taking public transportation”, “Worried about being infected by COVID-19 for yourself and your family”, “Whether to wear a mask”, “The impact of the epidemic on household income” (No effect/Increase/Decrease), and “The impact of the pandemic on family relations” (No effect/Positively/Negatively) being surveyed.

### 2.4. Generalized Anxiety Disorder 7-Item (GAD-7) Scale

The Generalized Anxiety Disorder Scale was compiled by Spitzer [17] and is used to screen for generalized anxiety and assess symptom severity. It consists of seven items and is widely used to discern how long respondents are troubled by seven questions, including “difficult to relax” and “excessively worried about various problems” in the past two weeks. GAD-7 has proven to be a reliable tool for measuring anxiety [18]. The GAD-7 scale is a 4-level self-rating scale for seven items, with a total score of 21 points. In this study, the GAD-7 scale is a 7-item self-rating scale with four grades, with a total score of 21 points, <5 points indicate no anxiety symptoms, ≥5 points indicate anxiety symptoms, of which 5–9 points have mild anxiety tendency, 10–13 is divided into moderate anxiety tendency, ≥14 is allotted into severe anxiety tendency.

### 2.5. Patient Health Questionnaire-9 (PHQ-9)

The Patient Health Questionnaire-9 (PHQ-9) is a questionnaire wielded to diagnose and determine the severity of depression. PHQ-9 is a patient self-report measurement that combines simplicity with “structure and standard validity” [19]. The Patient Health Questionnaire-9 (PHQ-9) uses a 4-level score (0–3), with a total score of 27 points. A total score of <5 points for the absence of depressive symptoms, ≥5 points for the presence of depressive symptoms, of which 5–9 points are mild depressive tendency, 10–14 points are moderate depressive tendency, 15–19 points are moderately severe depressive tendency, and 20–27 points were delimited as severe depression.

### 2.6. Impact of Event Scale (IES-R)

The IES-R is a self-administered questionnaire that has been well-validated in the Chinese population for deducing the extent of psychological impact after exposure to a public health crisis within one week of exposure [20]. The total IES-R score was divided into 0–23 (normal), 24–32 (mild psychological impact), 33–36 (moderate psychological impact), and >37 (severe psychological impact).

### 2.7. Statistical Analysis

A chi-square test and a Wilcoxon Rank Sum Test were utilized to compare the characteristics of the distribution for self-reported anxiety, depression, and PTSD symptoms. Order logistic regression analysis was conducted to screen the factors associated with self-reported anxiety, depression, and PTSD symptoms, and calculate the ORs (odds ratios) and a 95% CI (confidence interval). Non-conditional logistic regression was utilized to explore factors associated with PTSD symptoms among cases with anxiety or depression symptoms. Statistical tests were two-tailed with *p* < 0.05. The database was constructed with EpiDate 3.1 and analyzed with SPSS 25.0.

## 3. Results

### 3.1. The Prevalence of Anxiety, Depression, and PTSD Symptoms among College Students

Table 1 demonstrates the prevalence rates of anxiety, depression, and PTSD symptoms among college students. A total of 4754 canvasses completed the survey, 54.0% of respondents were female and the median age was 20 years. The self-reported rates of anxiety, depression, PTSD symptoms, any one, any two, and all three were 25.0%, 29.7%, 3.4%, 15.3%, 17.1%, and 2.9%, respectively. Most of the subjects with symptoms were mild and moderate. There were significant gender differences in the prevalence of anxiety symptoms, depression symptoms, and psychological symptom comorbidities (*p* < 0.05).

### 3.2. Distributions of Anxiety, Depression, and PTSD Symptoms among College Students

The distributions of anxiety, depression, and PTSD symptoms among college students are illustrated in Table 2. Among investigated college students, a significantly higher prevalence of self-reported anxiety symptoms was found in female cases (27.6% vs. 22.0%), those with higher educational background (31.0% vs. 24.7%), those who caught a cold recently (31.6% vs. 24.2%), those fearing another pandemic (32.2% vs. 18.4%), those fearing taking public transport (45.7% vs. 23.6%), those fearing contracting COVID-19 (31.7% vs. 17.3%), those who do not wear masks in public (30.0% vs. 24.4%), those with medium and above self-reported household income (27.8% vs. 23.1%), those with diminished household income due to COVID-19 (27.0% vs. 20.7%), those with worsened family relationships over COVID-19 (46.1% vs. 23.7%); a considerably higher prevalence of self-reported depressive symptoms was observed in female cases (31.5% vs. 27.6%), those with higher educational background (35.9% vs. 29.4%), those who caught a cold recently (37.2% vs. 28.8%), those fearing another pandemic (36.7% vs. 23.2%), those fearing taking public transport (51.3% vs. 28.2%), those fearing contracting COVID-19 (36.7% vs. 21.7%), those who do not wear masks in public (36.1% vs. 28.9%), those with medium and above self-reported household income (34.2% vs. 26.7%), those with decreased household income due to COVID-19 (32.0% vs. 24.6%), those with worsened family relationships over COVID-19 (53.2% vs. 28.2%); a quite higher prevalence of self-reported PTSD symptoms was found in those household registration in rural areas (3.9% vs. 2.6%), those with higher educational background (6.9% vs. 3.2%), those who caught a cold recently (5.2% vs. 3.1%), those fearing another pandemic (4.2% vs. 2.6%), those fearing taking public transport (11.5% vs. 2.8%), those fearing contracting COVID-19 (4.1% vs. 2.5%), those who do not wear masks in public (5.0% vs. 3.2%), those with medium and above self-reported household income (4.1% vs. 2.8%), and those with worsened family relationships over COVID-19 (10.6% vs. 2.9%).

### 3.3. Correlators Associated with Anxiety, Depression, and PTSD among College Students Stratified by Gender

Ordered logistic regression stratified by gender shows that fearing another pandemic, fearing contracting COVID-19, not wearing a mask in public, and deterioration of family relationships by COVID-19 were associated with anxiety and depression for both male and female college students (Table 3). In addition, having caught a cold recently and being affected by decreased household income due to COVID-19 were especially associated with anxiety symptoms for male college students; fear of taking public and self-reported low household income were especially associated with anxiety symptoms for female college students. Having caught a cold recently, fear of another pandemic, fear of taking public transport, fear of contracting COVID-19, not wearing a mask in public, self-reported low household income, and degeneration of family relationships due to COVID-19 were associated with depression symptoms for both male and female college students. In addition, higher educational background and deterioration of family relationships due to COVID-19 were particularly associated with depression symptoms for male college students. Fear of taking public transport and deterioration of family relationships were associated with PTSD symptoms for both male and female college students. In addition, rural household register, higher educational background, fear of another pandemic, and not wearing a mask in public were especially associated with PTSD symptoms for male college students; other majors in university and fear of contracting COVID-19 were especially associated with PTSD symptoms for female college students. Having caught a cold recently, fear of another pandemic, fear of taking public transport, fear of contracting COVID-19, not wearing a mask in public, and deterioration of family relationships due to COVID-19 were found to be linked with a higher risk for more psychological comorbidities in both male and female college students. In addition, high educational background and decreased household income due to COVID-19 were associated with a higher risk of more psychological symptoms for male college students; self-reported low level household income was associated with a higher risk of psychological comorbidities for female college students.

### 3.4. Correlators Associated with PTSD among College Students with Anxiety or Depression Symptoms

Table 4 displays the factors associated with PTSD among college students with anxiety or depression symptoms. The detection rate of PTSD in cases with anxiety or depression symptoms was 9.11% (7.90–10.49%), including 11.03% (9.00–13.46%) in male students and 7.70% (6.10–9.67%) in female students. Logistic regression showed that female (OR, 0.61, 95% CI: 0.43–0.86), rural household register (OR, 1.58, 95% CI: 1.08–2.32), higher educational level (OR, 1.90, 95% CI: 1.08–3.36), fear of taking public transport (OR, 2.84, 95% CI: 1.81–4.47), and deterioration of family relationship due to COVID-19 (OR, 2.30, 95% CI: 1.44–3.67) were found to be equated with a higher risk of comorbidity with PTSD symptoms among cases with anxiety and depression symptoms.

Stratified analysis by gender showed that other university majors (OR, 0.49, 95% CI: 0.29–0.84), rural household register (OR, 1.73, 95% CI: 1.00–2.97), higher educational background (OR, 2.26, 95% CI: 1.01–5.05), fear of taking public transport (OR, 5.52, 95% CI: 2.67–11.41), and deterioration of family relationships due to COVID-19 (OR, 1.75, 95% CI: 0.86–3.57) were associated with PTSD in male respondents with anxiety or depression symptoms. Other majors in university (OR = 1.77, 95% CI: 1.07–2.91) having caught a cold recently (OR, 2.02, 95% CI: 1.10–3.71), fear of taking public transport (OR, 1.99, 95% CI: 1.08–3.66), and deterioration of family relationships correlated with COVID-19 (OR, 2.89, 95% CI: 1.53–5.47) were related to PTSD symptoms in female respondents with anxiety or depressive symptoms.

## 4. Discussion

This study investigated the common psychological status of college students in the post-COVID-19 era and explored relevant factors. Our findings in the pre- and post- COVID-19 era revealed that levels of anxiety, depression, and PTSD symptoms among college students remained relatively high in the post-pandemic era [16]. It is evident that the epidemic itself continues to affect the mental health status of college students, especially out of concern for the development of the epidemic and the possibility of infection. The deterioration of family relationships and reduced family economic level due to COVID-19 were significantly associated with the three symptoms. Furthermore, there were gender differences in the mental health status among college students, with females more likely to experience anxiety and depression symptoms, and males from rural areas and higher educational backgrounds showing a fear of public transport contributing to a higher risk of PTSD symptoms.

The prevalence of anxiety, depression, and PTSD symptoms reported in this study was 25.0%, 29.7%, and 3.4%, respectively. Compared to the outbreak period, all three still show a higher level. The prevalence rates in our study were lower than in a previous cross-sectional study of young adults in the United States, whose respondents reported symptoms of anxiety, depression, and PTSD in 45.4%, 43.3%, and 31.8%, respectively [21]. This may be due to that study having been conducted one month after the declaration of a state of emergency in the United States for COVID-19, when the epidemic was still progressing at a high level, whereas our study was conducted when the epidemic was largely under control. We observed that the prevalence of PTSD symptoms in college students was lower than the yields of studies performed during the COVID-19 outbreak, while anxiety and depressive symptoms were in line with or higher than them [22,23,24,25]. Prior research identified a higher prevalence of mental health symptoms reported by students than the overall population due to multiple concerns about academic delays, financial support, reduced social relationships, etc. [26]. A worldwide meta-analysis revealed that mental health symptoms were highly prevalent during the COVID-19 pandemic and varied among populations, with medical students suffering the most severe mental health symptoms, followed by general students [27]. It was implied that despite being in the post-pandemic period, college students were still more likely to withstand psychological symptoms due to the persistent impact of sporadic cases in addition to pressure from study and future employment. Moreover, we also compared the prevalence of common psychological conditions of other high-risk groups during this period including medical staff and COVID-19 patients, which suggested that these studies reported considerably higher rates than our study [28,29]. This may be explained by the fact that they were directly affected by the COVID-19 infection or were at greater risk of exposure resulting in a poorer psychological status than other groups.

Family relationships were significantly associated with anxiety, depression, and PTSD due to COVID-19. Factors associated with the three psychological symptoms were both overlapping and different. Among those factors, deteriorated family relationships due to COVID-19 were linked with all three psychological symptoms. Several studies had revealed that family functioning showed a direct influence on the mental health of college students [30]. During the pandemic, students spent a long time with their family members and suffered from the same passive stress. The relatively closed and single-family space provided conditions for the mutual venting of negative emotions and further led to the deterioration of family relationships [31]. It’s reported that the divorce rate and deterioration of parent-child relationships during the pandemic were higher than those in other periods [32]. The deterioration of family relationships has a strong and lasting impact on the psychological status of college students. Although college students had returned to school, this correlation still existed, suggesting that more attention should be paid to enhance the maintenance of family relationships during and post the epidemic.

Previous studies had found family economic level was significantly associated with college students’ mental health [33,34]. In this study, dwindling family income caused by COVID-19 increased the risk of anxiety symptoms and depression symptoms in male students. This might be related to the social role of and pressure on males. Self-reported low household income contributed to a higher risk of depression symptoms for both male and female students. Students with poor family income levels experienced the long-term pressure of family economic burdens, which was prone to inducing negative emotions such as low self-esteem and depression [35].

Consistent with some previous findings [36], this study found that factors associated with anxiety symptoms, depression symptoms, and PTSD symptoms also included concerns or worries about the development of the epidemic and being infected. Although in the post-epidemic era, the impact of the epidemic itself on psychological problems continued. The post-epidemic era was not an epidemic-free era; along with the rise of asymptomatic infections and unexpected regional sporadic cases, coupled with the increase in population mobility in the post-epidemic era, people could still suffer from the tremendous pressure of the epidemic, including college students.

However, there were also gender differences, which were similar to the results of previous studies [37,38]. Identifying gender differences in risk factors was essential for accurate intervention. Our study suggested that females were more likely to develop anxiety and depression symptoms than males consistent with prior research [39,40,41,42]. In addition, we found that a higher proportion of PTSD comorbidity was observed in male cases than in female cases with anxiety or depression symptoms. Previous studies in the general population had shown that females were about twice as likely to develop PTSD as males [43,44,45]. This may pertain to the fact that the subjects of this analysis were college students who had been identified as suffering from anxiety or depression symptoms. In addition, it can be indicated from the results of factor analysis stratified by gender that rural college students, higher educational backgrounds, and fear of taking public transport contributed to a higher risk of PTSD symptoms among male college students. Male college students were relatively active in social activities, may have more opportunities to take public transport, and would then be more likely to panic or fear of being infected when local sporadic cases were reported. However, it is indispensable to know that this research still has some limitations: Firstly, in this research, we used a non-probability sampling survey instead of a random sampling survey; additionally, there were extensive variables equated with psychological symptoms in college students and only information closely related to COVID-19 was collected in this study. Additionally, the results may be varied in different COVID-19 settings, so the extrapolation and comparison of the results need to be performed cautiously.

## 5. Conclusions

In contrast to severe acute respiratory syndrome (SARS) and Middle East respiratory syndrome (MERS), the COVID-19 epidemic lasted longer. In the pre- and post-COVID-19 era, within our study, the three symptoms involved, anxiety, depression, and PTSD symptoms, have been at high levels from the period of the epidemic outbreak. This result is informative for public health emergencies. Our research also demonstrated the considerable impact of low household economic levels, deteriorating family relationships, and gender on the three symptoms. As subjects of social development, college students have social attributes such as working and reproducing offspring, so targeted strategies should be developed for them. Therefore, colleges are obliged to strengthen daily mental health education for students, screening and providing targeted psychological counseling.

## Figures and Tables

**Table 1 brainsci-12-01553-t001:** Self-reported rates of anxiety, depression, and PTSD symptoms among college students *.

Psychological Symptoms	N	Prevalence (95% CI)	N	Male	N	Female
				Prevalence % (95% CI)		Prevalence % (95% CI)
Anxiety						
Normal	3564	75 (73.9–76)	1706	78 (76.3–79.7)	1858	72.4 (70.3–74.3)
Mild	1057	22.2 (21.2–23.3)	417	19.1 (17.5–20.8)	640	24.9 (23.2–26.7)
Moderate	87	1.8 (1.5–2.2)	39	1.8 (1.3–2.4)	48	1.9 (1.5–2.4)
Severe	31	0.7 (0.5–0.9)	16	0.7 (0.4–1.3)	15	0.6 (0.3- 1.1)
Extremely severe	15	0.3 (0.2–0.5)	8	0.4 (0.2–0.7)	7	0.3 (0.1–0.6) ***
Depression						
Normal	3341	70.3 (69–71.6)	1582	72.4 (70.6–74)	1759	68.5 (66.8–70.1)
Mild	1155	24.3 (23.2–25.5)	488	22.3 (20.7–24)	667	26 (24.6–27.4)
Moderate	181	3.8 (3.2–4.5)	82	3.8 (3–4.6)	99	3.9 (3.2–4.7)
Severe	60	1.3 (1–1.6)	25	1.1 (0.7–1.8)	35	1.4 (1–1.9)
Extremely severe	17	0.4 (0.2–0.6)	9	0.4 (0.2–0.9)	8	0.3 (0.2–0.6) **
PTSD						
Normal	4332	91.1 (90.4–91.8)	1983	90.7 (89.6–91.7)	2349	91.5 (90.3–92.5)
Mild	262	5.5 (5–6.1)	119	5.4 (4.6–6.4)	143	5.6 (4.7–6.6)
Moderate	46	1.0 (0.7–1.3)	25	1.1 (0.8–1.7)	21	0.8 (0.5–1.3)
Severe	114	2.4 (2.1–2.8)	59	2.7 (2.2–3.2)	55	2.1 (1.6–2.8)
Any						
Normal	3078	64.7 (63.5–66.0)	1473	67.4 (65.5–69.2)	1605	62.5 (60.2–64.7)
Any one	1676	35.3 (34.0–36.5)	713	32.6 (30.8–34.5)	963	37.5 (35.3–39.8) ***
Comorbidity						
Normal	3078	64.7 (63.4–66.1)	1473	67.4 (65.3–69.4)	1605	62.5 (60.7–64.2)
Only one	725	15.3 (14.4–16.1)	325	14.9 (13.5– 6.4)	400	15.6 (14.2–17)
Any two	815	17.1 (16.2–18.2)	321	14.7 (13.3–16.2)	494	19.2 (18.1–20.4)
All three	136	2.9 (2.4–3.4)	67	3.1 (2.5–3.8)	69	2.7 (2.2–3.3) ***

Note: “*” *p* < 0.05, “**” *p* < 0.01, “***” *p* < 0.001.

**Table 2 brainsci-12-01553-t002:** Distributions of anxiety, depression, and PTSD symptoms among college students.

Variable	Group	N	Anxiety Prevalence (95% CI)	Depression Prevalence (95% CI)	PTSD Prevalence (95% CI)
Gender	Male	2186	22.0 (20.4–23.6)	27.6 (25.9–29.4)	3.8 (3.1–4.7)
	Female	2568	27.6 (26.2–29.2) ***	31.5 (29.8–33.2) **	3.0 (2.3–3.8)
Age					
	<18 years	179	25.7 (21.0–31.0)	33.5 (26.9–40.9)	3.9 (1.8–8.1)
	≧18 years	4575	25.0 (23.7–26.4)	29.6 (28.1–31.1)	3.3 (2.8–4.0)
Major during university					
	Medical	2733	24.1 (22.7–25.6)	29.6 (27.9–31.4)	3.3 (2.7–4.1)
	Others	2021	26.3 (24.6–28.0)	29.9 (27.7–32.1)	3.5 (2.9–4.2)
Household register					
	Urban	2036	23.8 (22.1–25.6)	29.3 (27.4–31.2)	2.6 (2.0–3.4)
	Rural	2718	25.9 (24.3–27.7)	30.1 (28.6–31.6)	3.9 (3.2–4.9) *
Education level					
	Undergraduate and below	4509	24.7 (23.4–26.1)	29.4 (27.9–30.9)	3.2 (2.7–3.7)
	Graduate	245	31.0 (25.4–37.3) *	35.9 (30.3–41.9) *	6.9 (4.6–10.3) **
Caught a cold within one week					
	No	4238	24.2 (23.1–25.4)	28.8 (27.8–29.9)	3.1 (2.7–3.6)
	Yes	516	31.6 (27.5–36.0) ***	37.2 (33.9–40.6) ***	5.2 (3.8–7.2) *
Fear of another pandemic					
	No	2465	18.4 (16.3–20.6)	23.2 (21.5–25.1)	2.6 (1.9–3.4)
	Yes	2289	32.2 (30.6–33.9) ***	36.7 (34.8–38.7) ***	4.2 (3.4–5.2) **
Fear of taking public transport					
	No	4450	23.6 (22.3–25.0)	28.2 (26.9–29.6)	2.8 (2.3–3.4)
	Yes	304	45.7 (39.1–52.5) ***	51.3 (44.1–58.5) ***	11.5 (8.4–15.5) ***
Fear of contracting COVID-19					
	No	2208	17.3 (15.8–19.0)	21.7 (20.4–23.1)	2.5 (1.9–3.3)
	Yes	2546	31.7 (29.8–33.7) ***	36.7 (34.9–38.5) ***	4.1 (3.4–4.9) **
Wearing a mask in public					
	Yes	4217	24.4 (23.2–25.7)	28.9 (27.6–30.3)	3.2 (2.8–3.6)
	No	537	30.0 (25.6–34.7) **	36.1 (32.0–40.5) **	5.0 (3.4–7.5) *
Self-reported household income					
	Low-income level	2826	23.1 (21.6–24.8)	26.7 (24.9–28.6)	2.8 (2.2–3.6)
	Medium and above	1928	27.8 (25.2–30.5) ***	34.2 (31.7–36.7) ***	4.1 (3.4–5.1) *
COVID-19 on household income					
	No impact or increase	1471	20.7 (18.8–22.8)	24.6 (22.7–26.7)	2.9 (2.2–3.8)
	Decrease	3283	27.0 (25.7–28.3) ***	32.0 (30.3–33.7) ***	3.6 (3.0–4.3)
COVID-19 on family relationships					
	No impact	4472	23.7 (22.8–24.7)	28.2 (27.2–29.3)	2.9 (2.4–3.5)
	Deteriorate	282	46.1 (39.8–52.5) ***	53.2 (47.4–58.9) ***	10.6 (6.9–16.1) ***

Note: “*” *p* < 0.05, “**” *p* < 0.01, “***” *p* < 0.001.

**Table 3 brainsci-12-01553-t003:** Correlators associated with anxiety, depression, and PTSD symptoms among college students stratified by sex.

Variable	Group	Anxiety OR (95% CI)		Depression OR (95% CI)		PTSD OR (95% CI)		Any OR (95% CI)	
		Male	Female	Male	Female	Male	Female	Male	Female
Age	≥18 years	0.77 (0.48–1.24)	1.43 (0.84–2.43)	0.71 (0.45–1.11)	1.11 (0.69–1.77)	0.91 (0.44–1.88)	1.04 (0.47–2.33)	0.73 (0.48–1.11)	1.21 (0.78–1.88)
Major during university	Others	0.86 (0.69–1.06)	1.17 (0.98–1.41)	0.92 (0.75–1.12)	0.97 (0.81–1.16)	0.80 (0.59–1.09)	1.41 (1.06–1.88)	0.86 (0.72–1.04)	1.08 (0.91–1.27)
Household register	Rural	0.90 (0.72–1.13)	1.02 (0.84–1.24)	0.92 (0.75–1.13)	0.83 (0.69–1.00)	1.52 (1.09–2.11) *	1.27 (0.92–1.76)	0.96 (0.79–1.16)	0.90 (0.76–1.07)
Education level	Graduate	1.48 (0.92–2.38)	1.33 (0.92–1.91)	1.70 (1.11–2.63) *	1.24 (0.87–1.77)	2.07 (1.17–3.66) *	1.03 (0.55–1.93)	1.74 (1.16–2.62) **	1.36 (0.98–1.88)
Caught a cold within one week	Yes	1.61 (1.19–2.19) **	1.26 (0.96–1.65)	1.43 (1.07–1.91) *	1.43 (1.10–1.85) **	1.34 (0.87–2.07)	1.49 (0.99–2.23)	1.39 (1.05–1.82) *	1.39 (1.09–1.78) **
Fear of another pandemic	Yes	1.71 (1.32–2.23) ***	1.28 (1.02–1.59) *	1.44 (1.13–1.84) **	1.27 (1.03–1.57) *	1.54 (1.05–2.26) *	1.29 (0.89–1.86)	1.67 (1.33–2.10) ***	1.31 (1.08–1.60) **
Fear of taking public transport	Yes	1.34 (0.84–2.16)	2.19 (1.65–2.91) ***	1.80 (1.15–2.82) *	2.02 (1.53–2.66) ***	4.92 (3.00–8.07) ***	2.48 (1.70–3.60) ***	1.81 (1.17–2.79) **	2.17 (1.67–2.82) ***
Fear of contracting COVID-19	Yes	1.67 (1.28–2.19) ***	1.69 (1.35–2.13) ***	1.59 (1.24–2.04) ***	1.67 (1.35–2.08) ***	1.19 (0.80–1.76)	1.85 (1.25–2.73) **	1.60 (1.27–2.01) ***	1.72 (1.41–2.11) ***
Wearing a mask in public	No	1.57 (1.17–2.11) **	1.51 (1.12–2.02) **	1.72 (1.32–2.25) ***	1.45 (1.09–1.93) *	1.71 (1.15–2.55) **	1.18 (0.74–1.89)	1.72 (1.33–2.21) ***	1.43 (1.09–1.87) **
Self-reported household income	Low	1.04 (0.84–1.30)	1.36 (1.13–1.65) **	1.23 (1.01–1.50) *	1.50 (1.25–1.81) ***	1.18 (0.87–1.61)	1.31 (0.97–1.76)	1.13 (0.94–1.37)	1.48 (1.24–1.75) ***
COVID-19 on household income	Decrease	1.28 (1.00–1.64) *	1.01 (0.81–1.25)	1.30 (1.04–1.63) *	1.04 (0.85–1.28)	1.30 (0.90–1.86)	0.97 (0.68–1.40)	1.29 (1.05–1.59) *	1.06 (0.88–1.29)
COVID-19 on family relationships	Deteriorate	3.36 (2.31–4.89) ***	2.45 (1.75–3.44) ***	3.63 (2.54–5.18) ***	2.48 (1.78–3.44) ***	2.47 (1.53–4.00) ***	2.99 (1.93–4.63) ***	3.36 (2.38–4.74) ***	2.35 (1.71–3.23) ***

Note: “*” *p* < 0.05 “**” *p* < 0.01 “***” *p* < 0.001.

**Table 4 brainsci-12-01553-t004:** Correlators associated with PTSD among college students with anxiety or depression stratified by sex.

Variable	Group	Total ^†^		Male ^‡^		Female ^‡^	
		Crude OR (95% CI)	Adjusted OR (95% CI)	Crude OR (95% CI)	Adjusted OR (95% CI)	Crude OR (95% CI)	Adjusted OR (95% CI)
Sex	Female	0.67 (0.48–0.94) *	0.61 (0.43–0.86) **		-		-
Age	≥18 years	1.07 (0.46–2.52)	1.02 (0.42–2.44)	0.83 (0.32–2.20)	0.71 (0.26–1.98)	2.56 (0.34–19.01)	2.54 (0.33–19.31)
Major during university	Others	1.00 (0.71–1.40)	0.93 (0.65–1.32)	0.53 (0.32–0.88) *	0.49 (0.29–0.84) *	1.82 (1.13–2.93) *	1.77 (1.07–2.91) *
Household register	Rural	1.46 (1.03–2.07) *	1.58 (1.08–2.32) *	1.44 (0.88–2.35)	1.73 (1.00–2.97) *	1.51 (0.91–2.5)	1.43 (0.81–2.53)
Education level	Graduate	1.93 (1.12–3.33) *	1.90 (1.08–3.36) *	2.5 (1.18–5.26) *	2.26 (1.01–5.05) *	1.49 (0.66–3.4)	1.43 (0.60–3.37)
Caught a cold within one week	Yes	1.39 (0.89–2.18)	1.39 (0.88–2.21)	0.9 (0.45–1.82)	0.87 (0.42–1.84)	1.99 (1.1–3.59) *	2.02 (1.10–3.71) *
Fear of another pandemic	Yes	1.03 (0.73–1.45)	1.08 (0.71–1.66)	1.23 (0.76–1.97)	1.32 (0.70–2.47)	0.96 (0.58–1.59)	0.82 (0.45–1.49)
Fear of taking public transport	Yes	2.66 (1.75–4.04) ***	2.84 (1.81–4.47) ***	5.75 (2.94–11.26) ***	5.52 (2.67–11.41) ***	2.03 (1.16–3.58) *	1.99 (1.08–3.66) *
Fear of contracting COVID-19	Yes	0.97 (0.69–1.39)	0.85 (0.55–1.31)	1.03 (0.64–1.68)	0.75 (0.39–1.44)	1 (0.59–1.67)	0.95 (0.51–1.75)
Wearing a mask in public	No	1.29 (0.82–2.03)	1.11 (0.69–1.80)	1.27 (0.71–2.29)	1.27 (0.66–2.41)	1.15 (0.56–2.39)	0.99 (0.46–2.12)
Self-reported household income	Low	1.18 (0.84–1.65)	1.04 (0.73–1.48)	1.11 (0.69–1.78)	1.06 (0.64–1.77)	1.24 (0.77–1.99)	1.09 (0.65–1.84)
COVID-19 on household income	Decrease	0.99 (0.68–1.44)	0.77 (0.51–1.17)	0.89 (0.53–1.5)	0.74 (0.41–1.30)	1.13 (0.65–1.99)	0.86 (0.46–1.59)
COVID-19 on family relationships	Deteriorate	2.43 (1.56–3.78) ***	2.30 (1.44–3.67) ***	1.74 (0.91–3.32)	1.75 (0.86–3.57) ***	3.27 (1.78–5.99) ***	2.89 (1.53–5.47) ***

Note: “*” *p* < 0.05 “**” *p* < 0.01 “***” *p* < 0.001. ^†^ dependent variable: screening results of PTSD in college students with anxiety or depression; independent variables: sex, age, major during university, household register, education level, caught a cold within one week, fear of another pandemic, fear of taking public transport, fear of contracting COVID-19, wearing a mask in public, self-reported household income, COVID-19 on household income, COVID-19 on family relationships. ^‡^ dependent variable: screening results of PTSD in college students with anxiety or depression by gender; independent variables: age, major during university, household register, education level, caught a cold within one week, fear of another pandemic, fear of taking public transport, fear of contracting COVID-19, wearing a mask in public, self-reported household income, COVID-19 on household income, COVID-19 on family relationships.

## Data Availability

The raw data supporting the conclusions of this article will be made available by the authors, without undue reservation.

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
