# Peer review of "Anxiety, Depression, and PTSD among College Students in the Post-COVID-19 Era: A Cross-Sectional Study"

_brainsci, 2022, doi:10.3390/brainsci12111553_

Round 1
Reviewer 1 Report
Dear Authors
Congratulations on your research. It is developed in a clear and simple way, the results and the method followed are very well understood. However, I found the discussion short, your results are very interesting and I expected to find a more intense discussion. I think this difficulty comes from the fact that you have made a brief theoretical framework, which is great because it complies with the policy of the journals but limits the ability to express ideas. As a constructive idea, I believe that they could contribute in the theoretical framework a greater number of meta-analysis studies and systematic reviews. I think it would greatly enrich the research.
Reviewer 2 Report
Thank you for giving me the opportunity to review this manuscript
I think it is necessary to revise the manuscript.
1) Please describe the study design in the title, the abstract, and the method section. I think this study is a cross-sectional study.
2)Please describe any eligibility criteria of the participants more clearly. Did they have any previous history of mental/physical disorders? Did they take any psychotropic medications? Did they experience COVID-19 infections?
3) Please describe how sample size was arrived at. Please describe any strategies to handle missing data.
4) Please describe what is the outcomes, potential confounders, and effect modifiers in this study. Please describe the population, the exposure, the control, and the outcomes clearly by using the PECO format. I think it is difficult to show the influence of COVID-19 on depression, anxiety and PTSD in this study. That is because it seemed that the many confounders were not controlled.
4) Please give any unadjusted estimates and, if applicable, confounder-adjusted estimates and their precisions (e.g, 95% confidence interval). Pleas e make clear which confounders were adjusted and why they were included.
5) Please describe the novelty of this study after referencing previous studies related to this study. For example, there were so many studies to evaluate the influence of COVID-19 pandemic on depression, anxiety, and PTSD. Please discuss whether and how the results of this study were in line with any previous studies including the following ones.
a) Cindy H Liu et al. Factors associated with depression, anxiety, and PTSD symptomatology during the COVID-19 pandemic: clinical implications for U.S. youth adult mental health. Psychiatry Res. 2020 Aug;290:113172
b) Cenat JM et al. Plevalence of symptoms of depression, anxiety, insomnia, posttraumatic stress disorder, and psychological distress among populations affected by the COVID-19 pandemic: A systematic review and meta-analysis. Psychiatry Res. 2021 Jan;295:113599
c) Liy CH. Risk factors of depression, anxiety and PTSD symptoms in perinatal women during the COVID-19 pandemic. Psychiatry Res. 2021 Jan;295:113552
d) Luceno-Moreno L et al. Symptoms of posttraumatic stress, anxiety, depression, levels of resilience and burnout in Spanish Health Personnel during the COVID-19 pandemic. Int J Environ Res Public Health. 2020 Jul 30;17(15):5514
e) Zuo K et al. COVID-19 related symptoms of anxiety, depression, and PTSD among US adults. Psychiatry Res. 2021 Jul; 301: 113959.
f) Chew NWS et al. A multinational, multicentre study on the psychological outcomes and associated physical symptoms amongst healthcare workers during COVID-19 outbreak. Brain Behav Immun. 2020 Aug;88: 559-565.
g) Xu Z et al. Loneliness, depression, anxiety, and post-traumatic stress disorder among Chinese adults during COVID-19. A cross-sectional online survey. Plos one. 2021 Oct 21;16 (10): e0259012
h) Lee CM et al. Anxiety, PTSD and stressors in mental health during the initial peak of COVID-19 pandemic. Plos one. 2021 Jul 29;16(7): e0255013
i) Parker C et al. Depression, anxiety, and acute stress disorder among patients hospitalized with COVID-19: A prospective cohort study. J acad consult liaison psychiatry. 2021 Mar-Apr;62(2): 211-219
j) Selsuk EB et al. Anxiety, depression, and post-traumatic stress disorder symptoms in adolescents during the COVID-19 outbreak and associated factors. Int J Clin Pract. 2021 Nov; 75(11): e14880.
k) Villarreal-Zegarra D et al. Depression, post-traumatic stress, anxiety, and fear of COVID-19 in the general population and health-care workers: prevalence, relationship, and explicative model in Peru. BMC psychiatry. 2021 Sep 17; 21(1):455
6) Please describe the aim and hypothesis of this study in the introduction.
I think it is necessary to revise the manuscript.
Reviewer 3 Report
The authors focused on important topic in psychiatry, such as “Anxiety, depression, and PTSD among college students in the post-COVID-19 era “.
Despite the fact that the text was written with the preservation of scientific manner , unfortunately references must be adopted to the journal guidelines.
Round 2
Reviewer 2 Report
Thank you for revising the manuscript.
This manuscript would be suitable for publication.